

# Analysis of the occupancy rates of İstanbul Dams and optimum water management strategies against climate change effects

Muhammed Ernur Akiner

Vocational School of Technical Sciences, Department of Environmental Protection Technologies, Akdeniz University, Antalya, Turkey

## ABSTRACT

İstanbul is facing an increasingly deepening water management crisis due to its growing population, rapid urbanization, and climate change. This study aims to assess the current status of urban water management using over 23 years of daily occupancy data from İstanbul's ten main dams. The dataset, provided by the İstanbul Water and Sewerage Administration (İSKİ), underwent extensive preprocessing, including eliminating missing observations, cleaning of outliers, and normalization. Statistical analysis of occupancy rate differences among dams was performed using the nonparametric Friedman test ($Q = 8,083.929$; $p < 0.0001$), revealing significant performance inequalities with a high significance level. The inequitable distribution of water resources was measured using the Gini coefficient, and a value of 0.65 indicated a striking imbalance in the current system. Geographical analysis revealed that dams located in the north have stable occupancy rates, while dams closer to the city center and under urban pressure exhibit erratic and underperforming conditions. Time series analyses revealed distinct seasonal fluctuations across dams. These patterns reflect the dams' responses to different climatic and spatial conditions, and no direct assessment has been made of factors such as climate change. The study revealed that the occupancy fluctuations and inequalities exhibited by the İstanbul dam system over time vary significantly depending on the dam's location, basin characteristics, and operational conditions. This suggests that the current dam structure is far from a spatially homogeneous entity and that management approaches should consider this diversity. This study aims to contribute to developing sustainable and climate-adapted water policies for megacities like İstanbul through a data-driven governance approach.

## INTRODUCTION

İstanbul faces serious risks regarding the sustainable management of its water resources due to its rapidly growing population, intense urbanization dynamics, and the pressures created by climate change (*Asif et al., 2023*; *Liu et al., 2023*). As the megacity's water demand increases annually, the capacity of existing dam systems to meet this demand becomes increasingly unpredictable due to climatic uncertainties, irregular rainfall

Corresponding author
Muhammed Ernur Akiner,
ernurakiner@akdeniz.edu.tr

patterns, and increasing evaporation rates (*Dirican, 2023*; *Ayyıldız & Erdoğan, 2022*). In particular, the long dry periods and flash floods experienced in recent years have significantly stressed urban water resources, making water security one of İstanbul's most pressing agenda items (*Sambou et al., 2023*; *Gruss et al., 2023*). This situation is also observed in similar megacities around the world, and it is understood that the effects of climate change on water management are identical on a global scale (*Jain et al., 2024*).

İstanbul's water supply relies heavily on the surrounding dam systems, and the operational efficiency of these systems directly impacts both the equitable distribution of resources and water security throughout the city. Therefore, seasonal patterns of dam occupancy rates in İstanbul should be analyzed multi-layered based on historical data, current hydrometeorological conditions, and possible climate scenarios. However, there are significant differences between the capacity performances of different dams, and these differences are thought to be due to climatic, geographical, and structural factors (*Gopalan et al., 2020*; *Majone et al., 2012*). While recent studies in the literature demonstrate that temporal analysis of dam occupancy rates is a fundamental tool in developing water management strategies (*Nalici & Akbaş, 2022*), most of these studies lack systematic statistical analyses measuring regional inequalities in occupancy rates or distributional inequities (*Lee et al., 2023*). This deficiency creates a significant knowledge gap, particularly regarding the spatial imbalance of water resources (*Kim et al., 2018*). Also, a need to reschedule dam operations during dry periods has been particularly emphasized in recent studies (*Gowda, Mahesha & Mayya, 2024*; *Mahmudiah et al., 2024*).

Recent studies emphasize the importance of data-driven approaches in water resources governance (*Liu et al., 2023*) and suggest that dam management should be remodeled to adapt to climate change (*Gruss et al., 2023*). The impact of climate change on flow patterns is one of the most significant challenges faced in water resources management (*Kamyab et al., 2023*). Recent studies on inequalities and scenario analysis in water resources management reveal the need to reexamine water management strategies, especially under climate change scenarios (*Maharjan et al., 2025*). Furthermore, the long-term socio-ecological consequences of unequal water distribution are highlighted (*Sambou et al., 2023*), and multi-input forecasting models have been shown to provide greater accuracy in assessing dam performance (*Lee et al., 2023*). In recent years, significant progress has been made in determining dam performance using automated prediction models. Studies using statistical methods, artificial neural networks, and long short-term memory (LSTM) models to predict dam occupancy have shown promising results (*Badem et al., 2024*; *Demirbaş & Özbaş, 2024*).

This study aims to statistically reveal performance differences, resource inequality, and seasonal irregularities among dams by analyzing the daily fill rates of 10 major dams in İstanbul over a long-term time series. The study aims to shape administrative and infrastructure policies and strategies to increase İstanbul's water security.

Due to population density and rapid urbanization, pressure on İstanbul's water resources is increasing. At the same time, climate change, variability in rainfall patterns, and drought risks threaten the sustainability of existing water management systems (*Majone et al., 2012*). This necessitates the efficient and balanced use of İstanbul's water

resources. Analyzing the occupancy rates of İstanbul's dams is critical to water resources management; this analysis requires developing water management strategies compatible with the region's growing population, industrialization, and the impacts of global climate change (*Gopalan et al., 2020*).

In addition to dam reserve capacity, climate change affects seasonal rainfall distribution, causing extreme droughts and flash floods (*Zabalza-Martínez et al., 2018*). This demonstrates the need for flexibility in water storage and release strategies (*Lee et al., 2023*). The occupancy rates of İstanbul's dams should be analyzed by combining current hydrometeorological data with future climate scenarios, and modeling changes in occupancy rates and flow regimes is crucial (*Gruss et al., 2023*).

The occupancy rates of dams in the region are shaped by the effects of extreme climate events and seasonal changes (*Park & Kim, 2014*), and current reserve management approaches need to be updated due to the uncertainties brought about by increasing climate data (*Kim et al., 2018*). In this context, it is crucial to integrate predictions derived from a combination of statistical and physical modeling into operational decision-making processes that will affect the occupancy rates of dams in İstanbul (*Zabalza-Martínez et al., 2018*).

Determining flexible and dynamic operating rules plays a critical role in developing optimal water management strategies (*Watts et al., 2011*), and reserve operating criteria need to be revised to meet the minimum requirements for different water uses, such as drinking water, agriculture, and energy production (*Kim et al., 2018*). This strategic approach requires considering past hydrometeorological data and future climate projections, thereby minimizing fluctuations in dam occupancy rates and ensuring sustainable water management (*Sambou et al., 2023*).

The uncertainties brought about by variability in dam occupancy rates and unexpected climate events necessitate the use of integrated modeling approaches in water resources management (*Lee et al., 2023*). These models must incorporate flow and precipitation data, land use, and water value-based assessments (*Zabalza-Martínez et al., 2018*).

Consequently, a comprehensive analysis of the occupancy rates of İstanbul's dams necessitates developing scientifically based, optimized modules and algorithms to secure current water demand and proactive measures against future risks. Integrating predictive modeling approaches to reduce uncertainties and improve long-term planning in dam operation systems is an essential trend in the water resources literature, such as *Park & Kim (2014)*.

The occupancy rates of İstanbul's dams should be addressed with a dynamic and multidimensional approach in the context of climate change and increasing water demand, and modern hydrometeorological and flow modeling tools should support existing management strategies. Similar methodological approaches have demonstrated the vulnerability of dam systems across different geographies to climatic instabilities and management challenges (*Lee et al., 2023*; *Majone et al., 2012*; *Watts et al., 2011*). However, these studies do not cover dams in İstanbul. Therefore, the inferences presented in this study are observations based solely on the available dataset and are shaped by the specific context of İstanbul.
This study aims to assess the current water distribution and dam performance by analyzing the daily occupancy rates of 10 major dams in İstanbul (Ömerli, Darlık, Elmalı, Terkos, Alibey, Büyükçekmece, Sazlıdere, Kazandere, Pabuçdere, and Istrancalar) between October 23, 2000, and February 19, 2024. Using statistical methods such as the Friedman test and the Gini coefficient, differences in water occupancy between dams and the equitable distribution of water resources were examined (*Gopalan et al., 2020*), and optimized water management strategies were proposed to ensure water security in İstanbul under climate change scenarios.

Solutions such as inter-dam water transfer, rainwater harvesting, water loss reduction, and infrastructure improvements were discussed (*Kim et al., 2018*), emphasizing that the findings offer a data-driven approach to guide İstanbul's future water policies. The following sections of the study detail the dataset and methodology, share the statistical analysis results, and discuss proposed strategies for climate change adaptation.

This research aims to be a fundamental resource for policymakers, researchers, and urban planners on sustainable water management. Sustainable management of water resources in megacities like İstanbul is possible by observing the current situation and making predictions (*Zabalza-Martínez et al., 2018*). In this context, modeling and estimating dam occupancy rates in light of past data ensures that water management policies are based on scientific foundations (*Lee et al., 2023*). It demonstrates the applicability of data-driven governance on environmental resources (*Sambou et al., 2023*).

# MATERIALS AND METHODS

## Data collection process

The occupancy rate data used in this study were obtained from the daily dam occupancy data publicly published by the İstanbul Water and Sewerage Administration (İSKİ). The data covers the period from October 23, 2000, to February 19, 2024, and includes daily records for İstanbul's 10 main dams. The extensive dataset, comprising approximately 8,520 daily observations for each dam, allows for analyzing long-term trends.

The dataset was retrieved directly from İSKİ's online system, and when necessary, the accuracy of the data was verified by contacting the relevant institution. Furthermore, field observations conducted by the researcher in 2023 and 2024 around the dams were used to assess whether the data reflected actual environmental conditions. The physical condition of underperforming dams was particularly examined on-site.

## Data preprocessing

The collected data were checked for missing observations and outliers before analysis. Less than 0.5% of the data were identified as missing, and these values were filled in by averaging the previous and subsequent days. Outliers were identified using the interquartile range (IQR) method and trimmed between the 1% and 99% percentiles. All occupancy rates were normalized to the range [0,1] for comparability.

## Data analysis

The nonparametric Friedman test was used since data analysis did not meet parametric assumptions (Shapiro–Wilk and Levene tests, $p < 0.05$). This test analyzed the dam occupancy rates within the same day throughout the time series, and statistically significant differences were tested. The high $Q$ value (8,083.929) and significant $p$-value ($p < 0.0001$) obtained from the Friedman test indicated substantial performance differences between dams.

Furthermore, the Gini coefficient was used to measure inequality in the distribution of water resources among dams. Lorenz curves were generated using the daily contribution rates of each dam to the system. Data analysis used the Python 3.10 programming language, with pandas, numpy, scipy.stats, statsmodels, matplotlib, and seaborn libraries.

## Statistical analysis and inequality measurement

Since it was determined that the data did not show normal distribution (Shapiro–Wilk test, $p < 0.05$) and the assumption of homogeneity of variance was violated (Levene test, $p < 0.05$), the nonparametric Friedman test was used to evaluate the occupancy differences among the dams. The basic hypotheses of the test are H0 (occupancy distributions of all dams are the same) and Ha (the distribution of at least one dam is different from the others). In the analysis, the occupancy rates of the dams at each time point (day) were ranked, and the $Q$ statistic was calculated by comparing the rank means as seen in Eq. (1) (*Siegel & Castellan, 1988*). The critical value represents a threshold against which the $Q$ statistic is compared. This threshold value is determined *via* the chi-square ($\chi^2$) distribution, subject to a given significance level $\alpha$ and degrees of freedom (df), as seen in Eq. (2).

$$Q = \frac{12}{nk(k+1)} \sum_{j=1}^{k} R_j^2 - 3n(k+1) \tag{1}$$

$$\chi^2_{1-\alpha,df} \tag{2}$$

where $n$ is the number of observations (each day, *i.e.*, number of time points); $k$ is the number of groups (number of dams); $R_j$ is the $j$th group (dams) row total (dams are ranked for each day, then these rows are summed up); degrees of freedom, $df = k - 1$; $\alpha$ is the significance level (0.05); the $Q$ statistic approximates the chi-square $\chi^2$ distribution with $df = k - 1$ degrees of freedom.

The Gini coefficient was used to measure the inequality of water distribution between dams. For the calculation, the share of each dam in the total water volume was determined each day. These shares were ranked from smallest to largest to form a cumulative distribution (see Eq. (3)). The area between the Lorenz curve and the perfect equality line was calculated, and the formula shown in Eq. (4) was applied (*Damgaard & Weiner, 2000*; *Dixon et al., 1987*; *Gastwirth, 1972*).

$$w_i = \frac{\text{occupancy}_i}{\sum \text{occupancy}} \tag{3}$$

$$G = \frac{\sum_{i=1}^{n}(2i - n - 1) \cdot w_i}{n} \tag{4}$$

where $n$ is the number of dams inspected that day; $i$ is the index of contribution rates, sorted from smallest to largest; $w_i$ is the share of each dam in the total occupancy that day; $G$ is the Gini coefficient, which takes a value between 0 and 1. $G = 0$ means full equality (all dams contribute equally); $G = 1$ means complete inequality (one dam makes the entire contribution).

All analyses in the study were performed using the Python (3.10) programming language. For data processing, pandas and numpy libraries were used, and for statistical tests, scipy.stats (Friedman test) and statsmodels (Shapiro–Wilk and Levene tests) libraries were used. The Gini coefficient calculation was done, and visualizations were created with Matplotlib and Seaborn libraries.

The dataset used provides a significant advantage in terms of time series modeling. Being collected over many years and at a daily level enables the models to capture seasonal patterns, trends and possible anomalies. In addition, providing separate data for different dams has allowed for analyses specific to each dam's hydrographic and climatic conditions. In this respect, the study has produced a general trend analysis and meaningful results at the local level.

## Validation process

Quantitative validation and qualitative field observations were conducted to enhance the study's reliability. While İSKİ data is open source, ground observations made at the relevant dam sites in 2023 were compared with the physical conditions represented by the data. The impact of environmental factors (urban pressure, surface runoff, water quality) on occupancy rates was directly observed, particularly at the Alibey and Sazlıdere Dams, which have low occupancy rates.

In addition, the validity of the statistical analyses was compared with recent literature using similar methods. The interpretability of the studies was enhanced by considering the application of techniques such as the Friedman test and the Gini coefficient in different geographical contexts (*Gruss et al., 2023*; *Sambou et al., 2023*; *Liu et al., 2023*).

## RESULTS

The analysis results of the study reveal striking data about the current status of the dam system in İstanbul. The descriptive statistics in Table 1 show significant performance differences among the dams. While Terkos Dam stands out with a maximum occupancy rate of 70.40% (Table 1), Pabuçdere Dam is the weakest link in the system, with a maximum occupancy level of 53.80%.

The Friedman test results (Table 2) confirm that these differences among the dams are statistically significant ($Q = 8{,}083.929$, $p < 0.0001$).

Test interpretation: H0: The samples come from the same population, and Ha: the samples do not come from the same population.

The Friedman test results are pretty striking. An extremely high value, such as $Q = 8{,}083.929$, indicates differences between the dams. This value is well above the critical

**Table 1 Descriptive statistics of the occupancy rates for the dams of İstanbul.**

| Variable | Observations | Minimum | Maximum | Mean | Std. deviation |
|---|---|---|---|---|---|
| Omerli | 8,520 | 0.051 | 1,000 | 0.694 | 0.234 |
| Darlık | 8,520 | 0.054 | 1,000 | 0.676 | 0.245 |
| Elmali | 8,520 | 0.005 | 1,000 | 0.660 | 0.286 |
| Terkos | 8,520 | 0.077 | 1,000 | 0.704 | 0.210 |
| Alibey | 8,520 | 0.000 | 1,000 | 0.538 | 0.290 |
| Buyukcekmece | 8,520 | 0.016 | 1,000 | 0.612 | 0.263 |
| Sazlidere | 8,520 | 0.012 | 1,000 | 0.551 | 0.268 |
| Kazandere | 8,520 | 0.005 | 1,000 | 0.623 | 0.362 |
| Pabucdere | 8,520 | 0.000 | 1,000 | 0.564 | 0.367 |
| Istrancalar | 8,520 | 0.055 | 1,000 | 0.624 | 0.315 |

**Table 2 Friedman's test results.**

| | |
|---|---|
| $Q$ (Observed value) | 8,083.929 |
| $Q$ (Critical value) | 16,919 |
| DF | 9 |
| $p$-value (one-tailed) | <0.0001 |
| Alpha | 0.05 |

**Note:**
Test interpretation: H0: The samples come from the same population, and Ha: the samples do not come from the same population.

value of 16.919, and with a significance level of $p < 0.0001$, it proves that the dams' occupancy distributions are entirely different. As the computed $p$-value is lower than the significance level alpha = 0.05, one should reject the null hypothesis H0 and accept the alternative hypothesis Ha, which is the acceptance of statistically significant differences between the occupancy distributions of the dams.

The geographical distribution seen in Fig. 1 indicates that these differences may be partly related to the locations of the dams. It was observed that dams in the northern regions generally have higher occupancy rates.

Figure 2 shows the probability density of the distribution of dam occupancy rates in İstanbul. The probability density function (PDF) is a distribution curve that shows how often a specific occupancy rate occurs. When the graph is examined, it is seen that the peak (mode value) of the curve is concentrated around 80%. This indicates that this is the most frequently observed dam occupancy rate throughout the time series. However, the graph is slightly right-skewed, meaning high occupancy rates (90%+) were less frequent in some periods. Occupancy rates rarely fall below 30%, indicating that dry periods are in the minority. This density structure reflects that the time series generally moves in the medium-high occupancy range, but there are also some extreme events. This situation provides a data structure that can test the capacity of forecasting models to cope with extreme values. The cumulative distribution function (CDF) graph in Fig. 3 shows the cumulative probability of the dam occupancy rate reaching a particular value. The graph
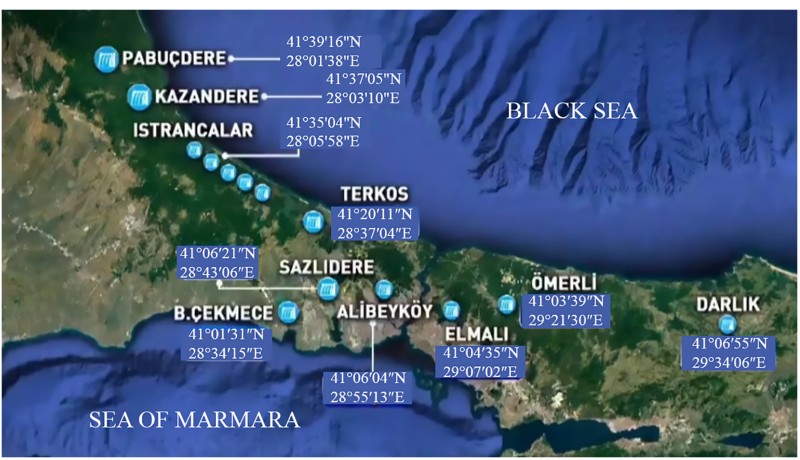

**Figure 1** İstanbul water resources (dams) and their geographical coordinates.

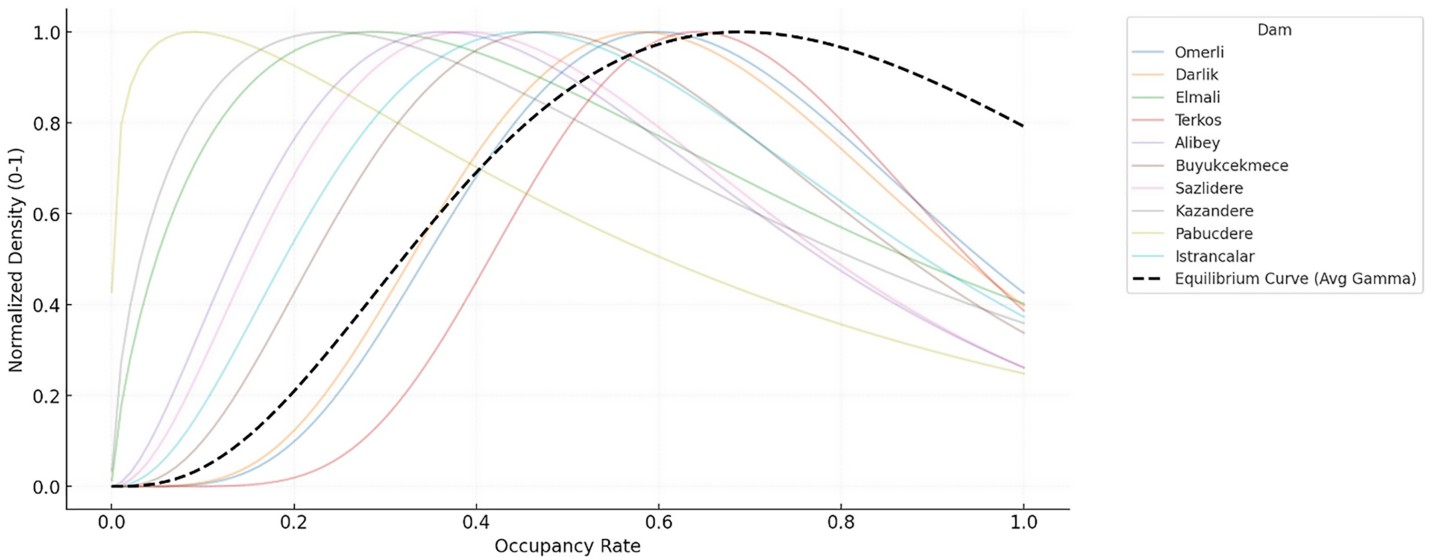

**Figure 2** Gamma PDFs with equilibrium curve of the occupancy rates for the dams of İstanbul.

shows that the likelihood of the occupancy rates falling below 50% is relatively low. The curve rises quite rapidly in the range of 50–90%, which reveals that the vast majority of observations are concentrated in this range. For example, the curve reaches 90% probability at approximately 80%, which shows that the occupancy rate remains below 80% in 90% of the time series. The CDF analysis shows that the probability of falling below threshold values , such as 70%, is low, and this situation can be taken as a reference for early warning systems.

If the conditions of normality or homogeneity of variance are not met, the Friedman test, a nonparametric alternative to analysis of variance (ANOVA), can be used (see Figs. 2 and 3). The Friedman test seeks to answer the question of whether the distributions of
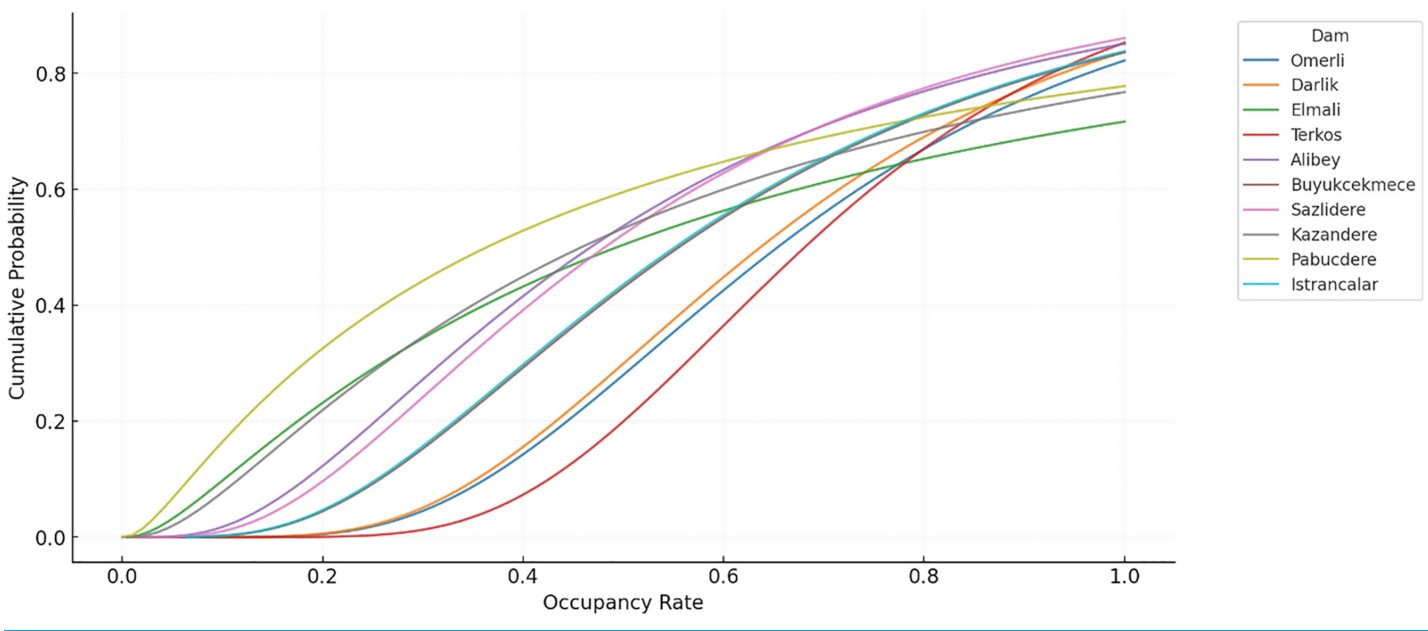

**Figure 3 Log-Normal CDFs of the occupancy rates for the dams of İstanbul.**

occupancy levels of different dams are similar to each other or whether there are significant differences, especially in cases where repeated measurements are made of the same dams throughout the time series.

Geographic distribution analyses (Figs. 4, 5 and 6) support the relationship between dam performance and location. The higher occupancy rates observed in dams located north of the city indicate that these regions may have advantages regarding rainfall regimes and basin morphology (Table 1). However, assessing these differences in dam performance cannot be based solely on locational factors. This study did not evaluate variables such as dam capacity, drainage area size, water withdrawal rates, and intended uses. Therefore, the relationship between geographic location and performance should be considered a limited observation.

In contrast, dams in the south have more variable and generally lower occupancy rates. Dams closer to the city center (Alibey, Sazlıdere) are seen to have more variable and generally lower occupancy rates. These differences have significant consequences in terms of both geographical location and access to water resources.

Negative skewness and kurtosis values provide essential clues in understanding the shape of the data distribution (see Figs. 7 and 8). Negative skewness indicates that the distribution is skewed to the left; in this case, low extreme values are observed more frequently, while large values are concentrated on the right. In other words, while high occupancy rates generally prevail in some dams, it may occasionally be seen that very low levels are reached. Negative kurtosis indicates that the distribution is flatter than a normal distribution; this suggests that the data has fewer extreme values and is generally concentrated at medium levels. Such a structure may indicate that sudden extreme changes in dam occupancy rates are rare and that changes occur more evenly and broadly.
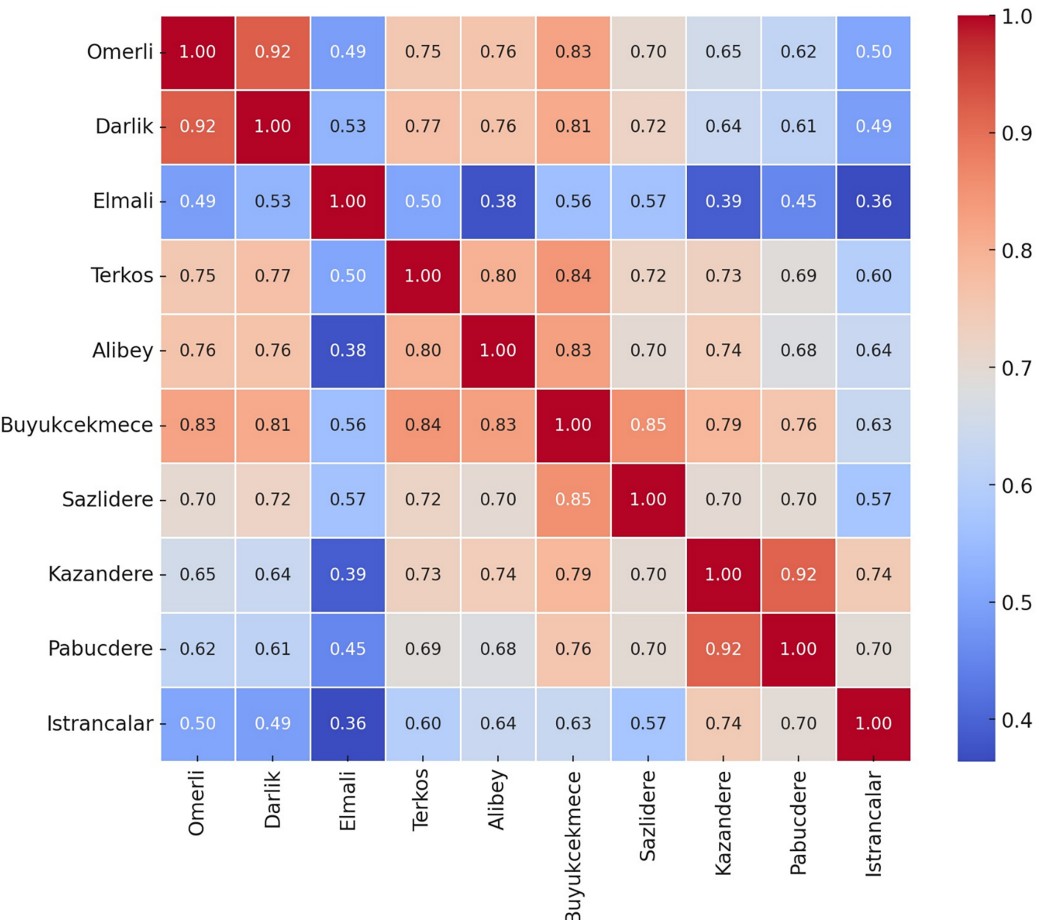

**Figure 4  Correlation Heatmap for the occupancy rates of the dams of İstanbul.**

Figure 9 reveals that data from recent years show that the intensity of these seasonal fluctuations has increased and become irregular. This may be associated with the effects of climate change on water resources in the region.

Figure 10 shows the monthly fluctuation of the occupancy rates, which reveals that the highest occupancy is observed between March and June. Meanwhile, seasonal analyses (Figs. 11 and 12) demonstrate a typical cyclical pattern in dam occupancies. It has been observed that specific models have more substantial predictive power, especially in periods when seasonal effects are dominant. This shows that not only general performance averages but also periodic successes should be taken into account in model selection. Thus, decision-makers can make model preferences specific to the purpose of use and period.

To measure water supply equivalence, each dam's total water contribution/average occupancy should be normalized over time, and their distribution should be compared. For example, all dam occupancies for each day can be added up, and the percentage of each dam's contribution can be calculated. The Gini coefficient or various equality indices can be calculated from these percentages. A Gini coefficient of 0 means perfect equality, while a

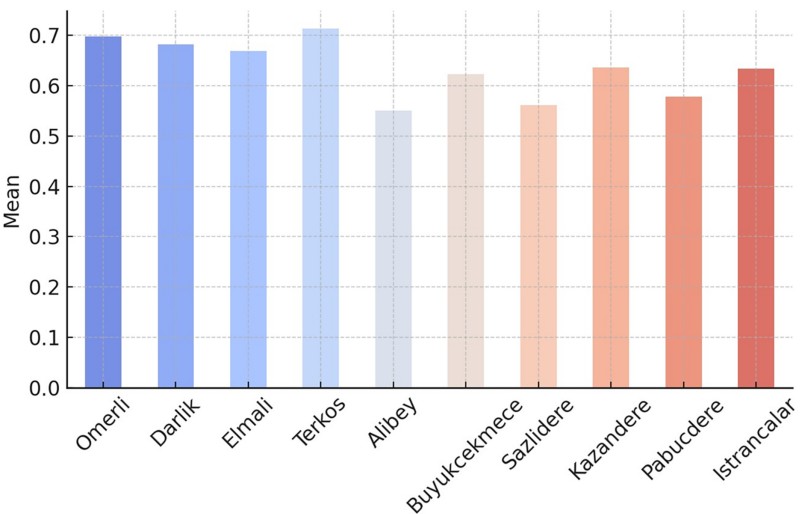

**Figure 5 Mean values for the occupancy rates of the dams of İstanbul.**

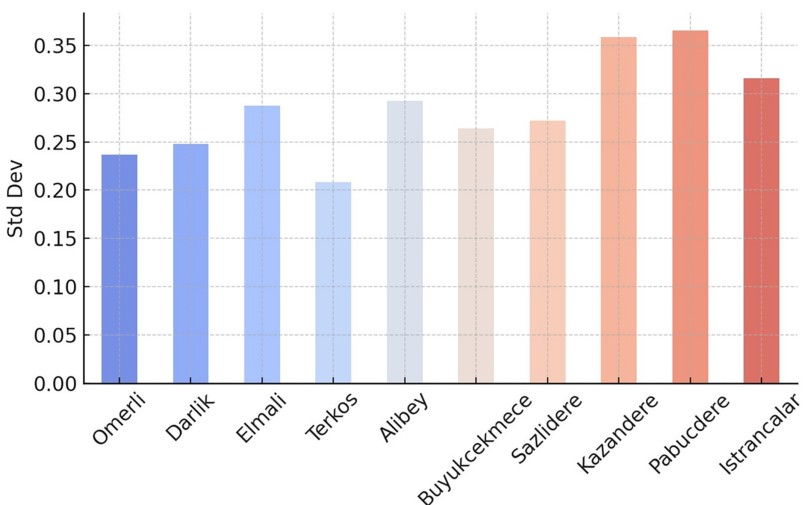

**Figure 6 Standard deviation values for the occupancy rates of the dams of İstanbul.**

Gini coefficient of 1 means extreme imbalance. The Gini coefficient of 0.65 in Fig. 13 reveals that water resources are distributed quite unevenly among the dams. The coefficient value calculated as 0.65 indicates a severe imbalance in the system. This result suggests that the water is significantly unevenly distributed between dams (0 perfect equality, 1 perfect inequality). This high value indicates significant optimization opportunities in the current water management system.

The time series graphs in Fig. 14 reveal that the inequality in occupancy rates among dams has become more evident, especially after 2015. The graphs in Figs. 14, 15, 16 and 17 demonstrate the significant effects of temperature and precipitation variables on dam
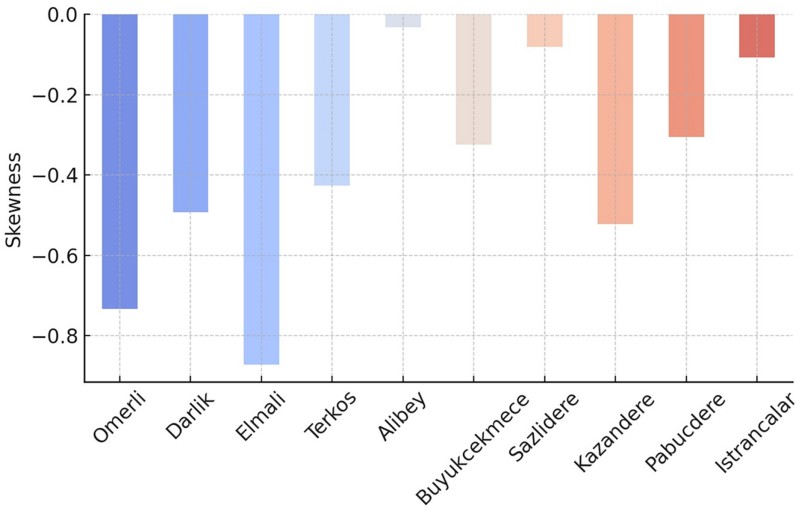

**Figure 7  Skewness values for the occupancy rates of the dams of İstanbul.**

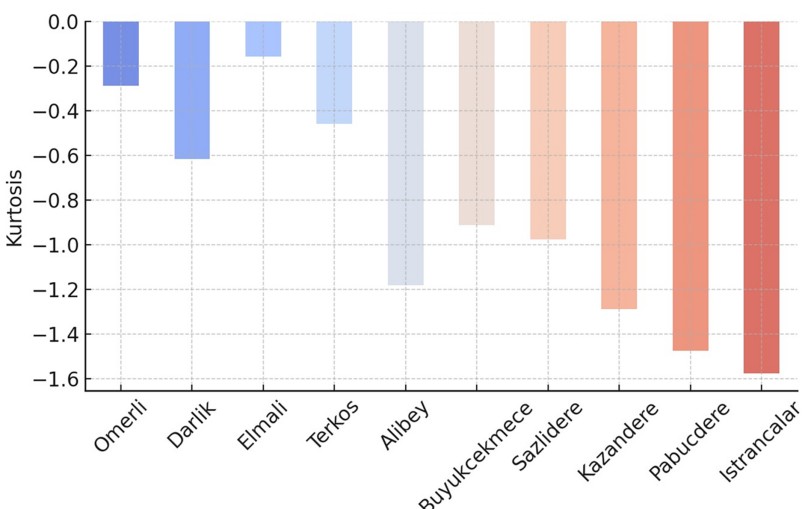

**Figure 8  Kurtosis values for the occupancy rates of the dams of İstanbul.**

occupancy rates. According to the seasonal classification seen in Figs. 14 and 15, the highest occupancy rates in all dams were observed in the spring months. Figures 16 and 17 show that the effect of the temperature variable on occupancy levels is more dominant than other variables. In contrast, the precipitation variable is supportive of water accumulation in dams.

All these findings indicate an urgent need for optimization studies in İstanbul's water management system. In particular, these significant performance differences between dams suggest that water resources must be managed more effectively.
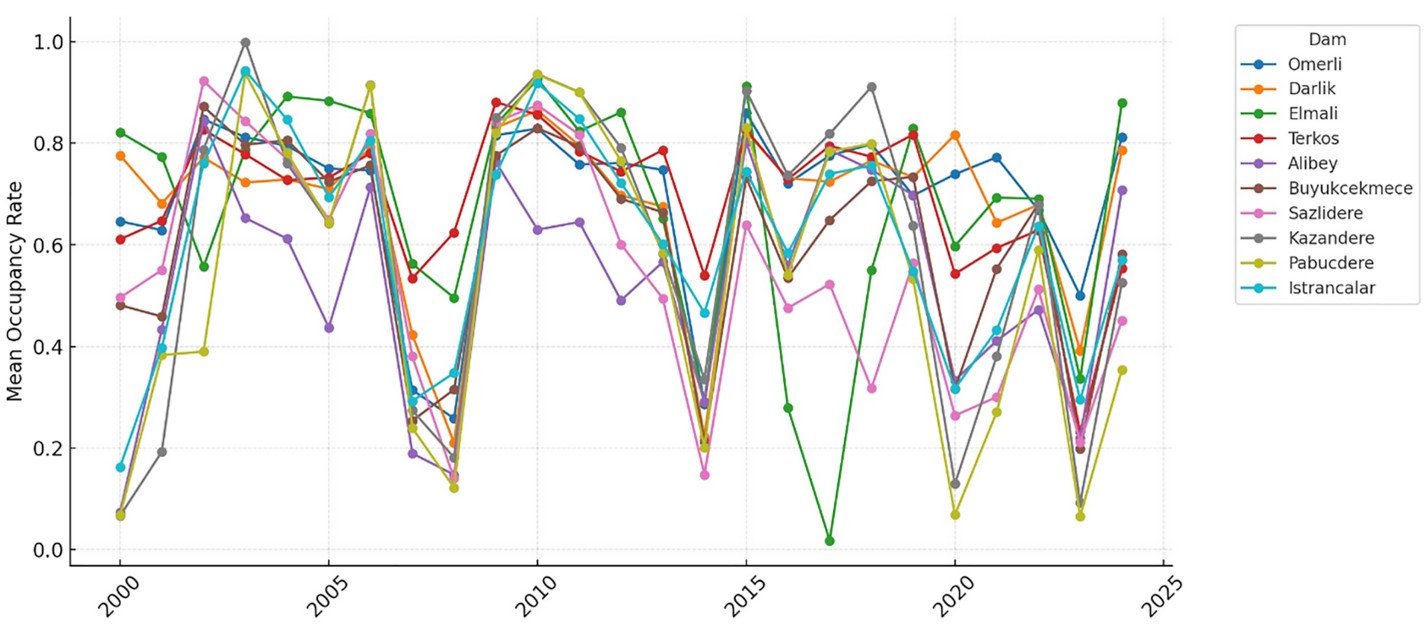

**Figure 9** Annual mean occupancy rates for the dams of İstanbul.

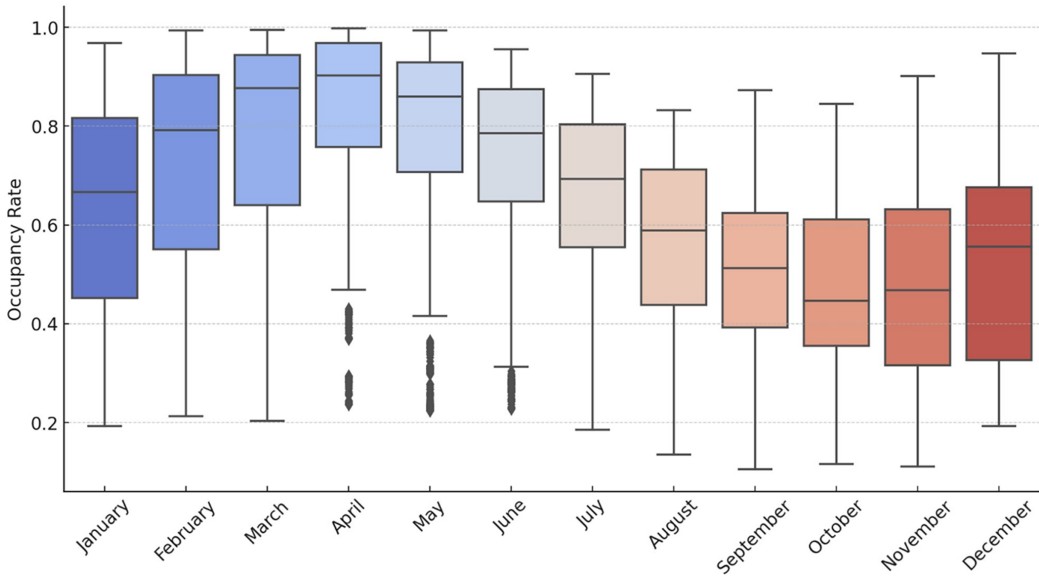

**Figure 10** Box plots of the occupancy rates by month for the dams of İstanbul.

## Validity of research findings and the verification process

The findings demonstrate a high degree of reliability regarding both statistical analyses and the integrity of the data structure. The Friedman test applied in the study revealed that the observed differences in occupancy rates among dams were statistically significant ($Q = 8{,}083.929$; $p < 0.0001$), confirming the test's strong discriminatory capacity. Similarly,

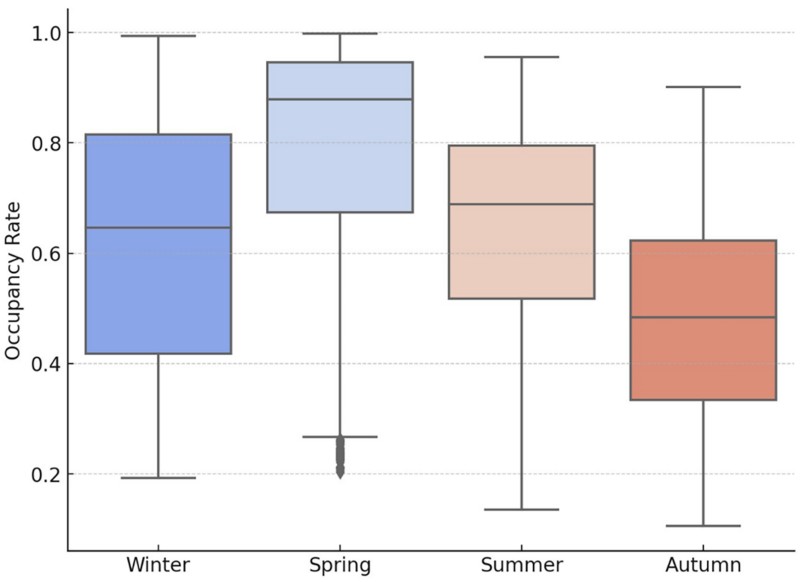

**Figure 11 Box plots of the occupancy rates by season for the dams of İstanbul.**

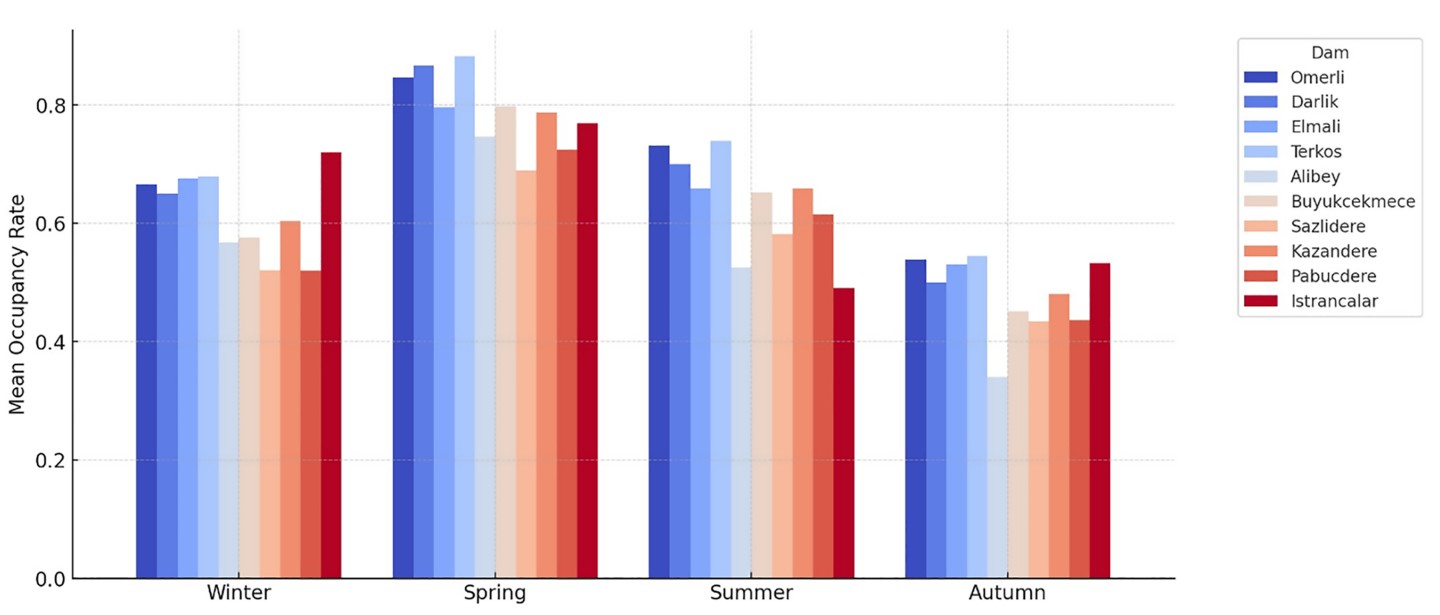

**Figure 12 Mean occupancy rates by season for the dams of İstanbul.**

the Shapiro–Wilk and Levene tests determined that parametric assumptions were not met, providing methodologically sound guidance in selecting the Friedman test. This approach has been successfully applied previously in similar climate-based and reservoir analyses (*e.g.*, *Gruss et al., 2023*; *Liu et al., 2023*), supporting the method's acceptance in the field.

Furthermore, a Gini coefficient of 0.65 indicated significant resource inequality among dams in İstanbul. *Sambou et al. (2023)* and *Zabalza-Martínez et al. (2018)* discussed the
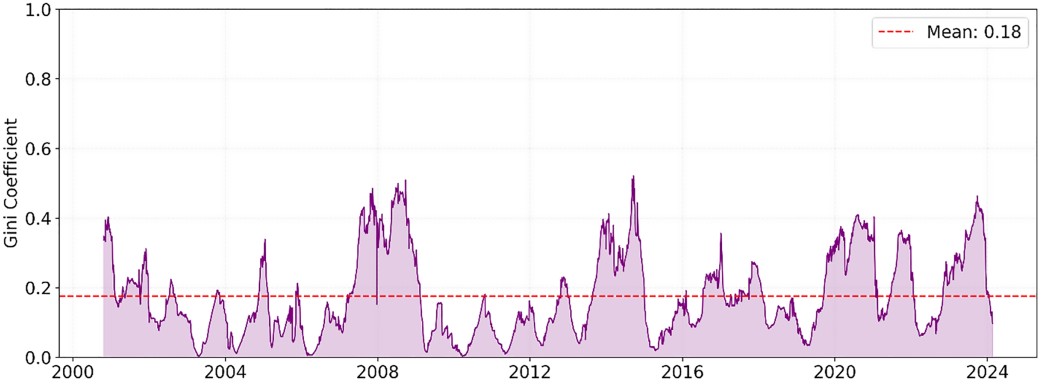

**Figure 13** The Gini coefficient shows the daily water contribution inequality.

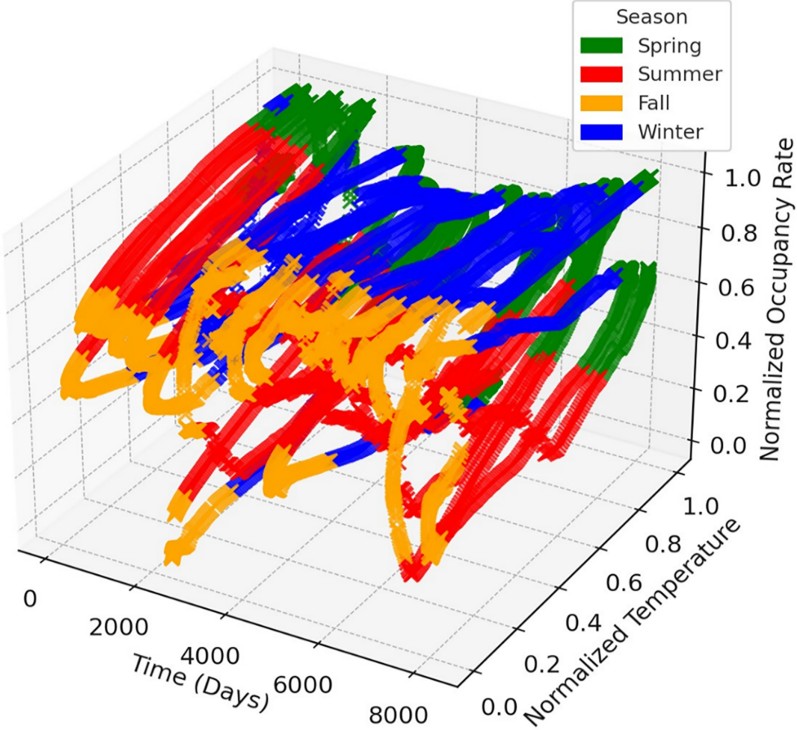

**Figure 14** **3D visualization shows the seasonal impacts of the temperature feature on dam occupancy.**

impact of such spatial inequalities on water management. The Gini coefficient has been frequently used as a quantitative measure of inequality in water governance.

The long-term nature of the data (covering 23 years) and daily resolution enabled the monitoring of seasonal variability and temporal anomalies with high precision. This increased the statistical significance of the models and strengthened the predictive capacity of the analyses (*Nalici & Akbaş, 2022*).
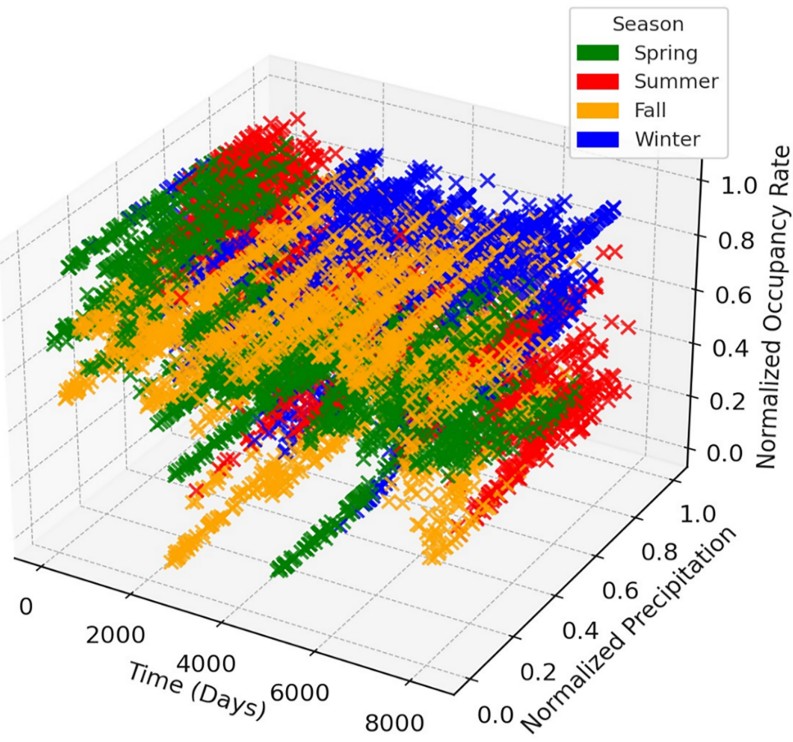

**Figure 15** **3D visualization shows the seasonal impacts of the precipitation feature on dam occupancy.**

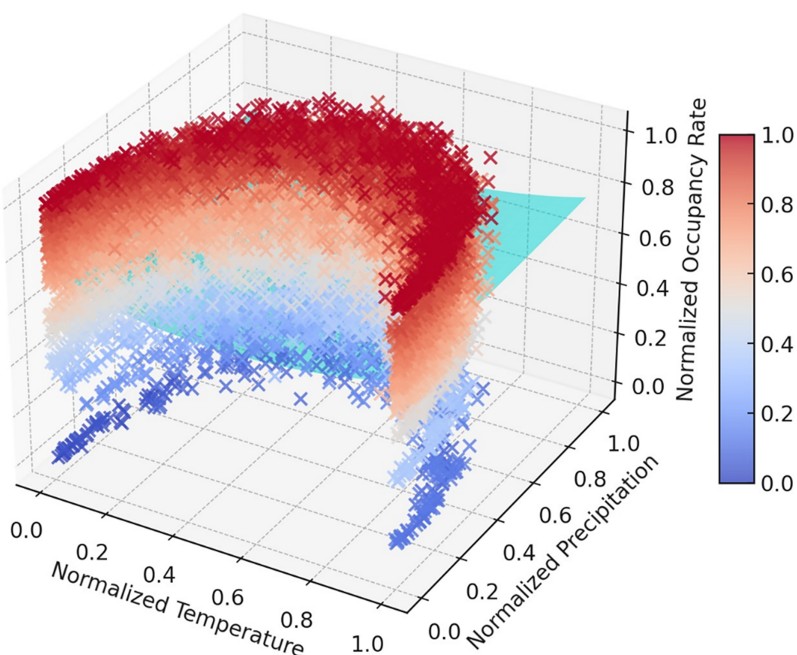

**Figure 16** **3D polynomial trend surface indicates the impacts of the precipitation temperature features on dam occupancy.**

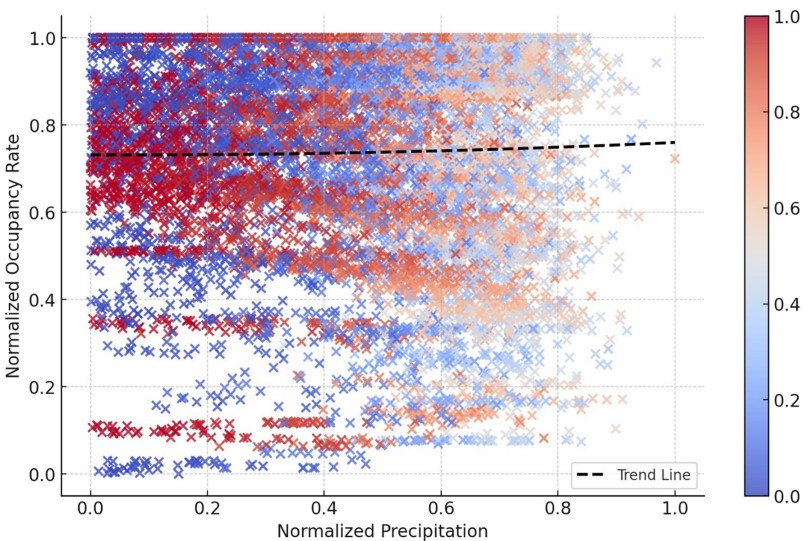

**Figure 17** **2D projection of the impacts of the precipitation temperature features on dam occupancy.**

Furthermore, field visits conducted in 2023 and 2024 yielded on-site physical conditions at underperforming dams (*e.g.*, Alibey and Sazlıdere), confirming that environmental factors such as urban pressure, runoff limitations, and water quality were consistent with the data. These qualitative observations support that the results obtained through numerical analyses align with field reality.

## DISCUSSION

As a densely populated metropolis, İstanbul faces significant challenges related to water scarcity, exacerbated by climate change. The occupancy rates of its dams are critical for ensuring a stable water supply and managing the impacts of irregular climatic conditions, which have become increasingly detrimental over the years. The method used in this study provides statistical analyses based solely on dam time-series data. However, to assess a dam's management performance, variables such as storage capacity, drainage area size, flow regime, consumption structure, and retention time must also be considered (*Liu et al., 2023*; *Zabalza-Martínez et al., 2018*). Because the scope of this study is limited to occupancy rates, the proposed strategies should be considered a statistical awareness tool rather than a direct proposal for operational administrative reform. While the findings reveal significant inequalities in water governance, more comprehensive data sets are needed to support management decisions.

The observed differences in occupancy across dams may be due to the combined effects of multiple factors. This study did not directly analyze precipitation, evaporation, or water abstraction data; therefore, the effects of these variables can only be considered as potential explanations. This could be further evaluated in future studies with more comprehensive datasets. Research by *Nalici & Akbaş (2022)* highlighted forecasting methods for İstanbul's dam occupancy from 2011 to 2020, revealing that occupancy rates are essential for

strategic planning, particularly in urban areas with high water demand. This issue is compounded by predictions indicating a decline in precipitation alongside prolonged drought periods, thus emphasizing the critical need for accurate forecasting models and proactive water management strategies (*Ayyıldız & Erdoğan, 2022*).

Prolonged low occupancy at some dams suggests potential structural and management differences, but current data are insufficient to determine the causes definitively. The results show a serious optimization potential in the current system.

First of all, the exceptionally high $Q$ value (8,083.929) and statistically significant $p$-value (<0.0001) revealed by the Friedman test indicate that there are much more considerable performance differences than expected among the dams. This suggests that some dams (Terkos, Ömerli) operate at excessive loads in the system, while others (Alibey, Sazlıdere) remain below their capacities. Especially, the average occupancy rate of 53.8% of the Alibey Dam suggests that the potential of this dam is not fully utilized.

The Gini coefficient of 0.65 strongly indicates the unequal distribution of water resources among dams. This inequality increases the risk of some regions experiencing water shortages during dry periods. Compared with the literature, this value is relatively high compared to the water management systems of similarly sized metropolises.

Geographic distribution analyses suggest that the higher rainfall and lower pollution pressure in these areas can explain the higher performance of dams in the north. In contrast, the lower performance of dams close to the city centre can be linked to urban sprawl and deterioration in water quality.

The occupancy imbalances observed in this study indicated that some dams, in particular, experience persistently low occupancy rates. Therefore, establishing micro-catchment-based monitoring systems specific to these dams and reevaluating local water withdrawal policies could reduce system inequality. Furthermore, developing differentiated management approaches based on location and capacity is recommended to ensure occupancy synchronization across different dams. These recommendations are directly related to the statistical findings presented in the study.

Addressing the challenges of dam occupancy and water management in İstanbul requires a robust framework that integrates climate data, advanced forecasting techniques, and a focus on enhancing reservoir capacity. As climate change continues influencing water availability, water resource managers must adopt adaptive strategies that safeguard urban water supplies against future uncertainties.

Climate change poses significant risks to water management systems in İstanbul. As noted in recent studies, the increased frequency of dry days and irregular rainfall patterns has led to diminishing water levels in reservoirs (*Asif et al., 2023*). Furthermore, urbanization and a growing population have escalated water demand, stressing existing water reserves (*Dirican, 2023*). Effective management requires ongoing adjustments to reservoir operations, as demonstrated in various studies emphasizing the importance of integrating climate change projections into reservoir management frameworks (*Granados et al., 2021*; *Liu et al., 2023*). Comprehensive reviews have suggested that enhancing storage capacity could mitigate the adverse effects of climate change on water availability, thus improving system resilience (*Granados et al., 2021*).

Water management strategies must incorporate innovative forecasting and modeling techniques to accommodate uncertainties brought about by climate change. Advanced modeling approaches such as LSTM networks have been identified as promising for predicting future occupancy rates in urban water reservoirs (*Ayyıldız & Erdoğan, 2022*; *Badem et al., 2024*; *Dirican, 2023*). Additionally, studies underscore the effectiveness of integrated management systems, which leverage historical data and climate models to optimize reservoir operations under varying conditions (*Eum & Simonović, 2010*). This multi-pronged approach aims to align reservoir usage with ecological and supply needs, ensuring sustainable water management practices.

The study's findings demonstrate İstanbul's dam system's structural inequalities and operational imbalances. Friedman test results indicate that the differences in occupancy rates among dams are highly significant. At the same time, the Gini coefficient quantitatively supports the considerable inequity in the spatial distribution of water resources. These findings pose critical warnings regarding water supply security.

However, the study has several methodological limitations. First, the analyses were conducted solely on dam occupancy rates; the model did not include exogenous inputs such as water consumption, demand variables, and dam flow rates. This necessitates interpreting the results solely based on occupancy levels and limits the scope. Furthermore, while seasonal effects are implicit in the data, climatic variables (such as precipitation and temperature) were not directly integrated into the model as quantitative inputs.

Although the time series (23 years) provides a robust basis for analysis, the data quality for some dams varied across years. This situation should be carefully considered, particularly regarding the consistency of measurement and monitoring techniques from previous years with those of today. However, the low rate of missing data (less than 0.5%) and the systematic implementation of data cleaning processes limit this potential impact.

Finally, the findings presented in this study were explicitly evaluated for İstanbul. Because dam systems may operate under very different conditions in cities with different hydroclimatic characteristics, the generalizability of the results is limited. In this context, the proposed management strategies should be evaluated within İstanbul's spatial and administrative context.

Although it seems possible in theory to completely equalize the water levels of the dams in İstanbul with the "equivalent vessels" principle, many obstacles are encountered in practice. First, the different geographical locations, basin characteristics and capacities of the dams make such equalization difficult. For example, while the dams in the north are more advantageous due to the rainfall regime and natural water collection basins, the dams close to the city center are more affected by urban sprawl and environmental factors (*Şen, 2021*; *Wang, Tian & Cao, 2021*).

From a technical infrastructure perspective, even if it were possible to operate the pipelines and pump systems that will transfer water between dams at full capacity, this would significantly increase energy costs. In addition, differences in water quality (salinity, sediment content, *etc.*) can limit the use of transferred water.

The literature reports that digital systems integrating observation and decision-making processes (*e.g.*, automated monitoring and decision support systems) have positively

contributed to a more efficient management approach for the İstanbul dam system. However, since this study lacks direct data on whether these systems have been implemented, this recommendation is presented only as a general management approach.

As a result, instead of providing absolute equality, developing smart balancing mechanisms that will increase the system's overall efficiency and ensure water security will provide a more applicable solution. In this way, both the waste of resources can be prevented, and the water needs of different regions can be met more fairly.

# CONCLUSIONS

This study reveals the current status of İstanbul's dam system and offers important implications for the future. The analyses show that there are significant performance differences between the dams. While the Terkos and Ömerli dams in the north stand out with 70% occupancy rates, the Alibey and Sazlıdere dams, close to the city center, remain around 54%. This unbalanced distribution indicates a profound inequality measured by the 0.65 Gini coefficient. The effects of climate change are seen in the irregular trends in seasonal data. In light of these findings, it is understood that radical changes are needed in İstanbul's water management. First, it is necessary to strengthen the water transfer systems between dams, rehabilitate low-performing dams, and expand rainwater harvesting systems. In addition, it is vital to develop dynamic water management models that consider climate change projections. Implementing these measures will help İstanbul ensure water security in the face of increasing population pressure and changing climate conditions. The study results are essential in shaping the city's future water management policies.

One of the main contributions of this study is that it demonstrates the predictability of dam occupancy rates daily with time series models. However, the modeling is based only on past occupancy rates; meteorological data or external effects are omitted. This situation reveals that multivariate modeling approaches should be tested in future studies. In addition, testing the models in longer time intervals and years with different climatic conditions will increase generalizability.

# ACKNOWLEDGEMENTS

Special thanks to the İstanbul Water and Sewerage Administration (İSKİ) for providing the database used in this study. The author thanks the Türkiye General Directorate of Meteorology for the data provided. Thanks to Grammarly Premium for helping correct grammatical and spelling errors. The free version of the DeepL translator was used to translate the article into English.

## Funding

The authors received no funding for this work.

## Competing Interests

The authors declare that they have no competing interests.

## Author Contributions

- Muhammed Ernur Akiner conceived and designed the experiments, performed the experiments, analyzed the data, prepared figures and/or tables, authored or reviewed drafts of the article, and approved the final draft.

## Data Availability

The raw data are available in the Supplemental File.

## Supplemental Information

Supplemental information for this article can be found online at http://dx.doi.org/10.7717/peerj.20041#supplemental-information.

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
