# Peer review of "Analysis of the occupancy rates of İstanbul Dams and optimum water management strategies against climate change effects"

_PeerJ, doi:10.7717/peerj.20041_

## Round 0.1 · original submission · Major Revisions

Dear Muhammed Ernur Akiner,

Thank you for submitting your manuscript entitled: "Analysis of the occupancy rates of İstanbul Dams and optimum water management strategies against climate change effects." The reviewers have now completed their evaluations. After carefully considering their comments and suggestions, I agree that your study addresses a timely and important topic with significant implications for water resource management in the context of climate change.

However, reviewers have recommended major revisions before the manuscript can be considered for publication.

You are therefore invited to submit a thoroughly revised version of your manuscript that fully addresses the reviewers’ concerns. Please include a detailed, point-by-point response letter outlining how each comment has been considered and incorporated into the revision.

We appreciate your interest in publishing with PeerJ and look forward to receiving your revised submission.

Best regards,
Dr. Armando Sunny

·

Basic reporting

The abstract needs to be refined to include all essential components: study context, problem statement, methodology, key findings, and significance.
I suggest that Abstract should be written without headings, and it should have key words at the end.
Keywords should accurately reflect the study's content. In this case key words are missing at all.
The introduction requires significant improvement. Add a clear overview and cite recent and relevant studies to contextualize your research.
-Add the literature review to include recent publications (within the last 2-3 years)

Experimental design

1. Methodology Presentation

I appreciate the methodology part but one thing I couldn’t clearly find was validation through field visits. No study can be completed without filed visits, so I suggest it should be clearly indicated.
There is no mention of data collection, data analysis and validation of research processes, please use sperate headings and then present your arguments.
2. Model Development and Validation
• Clearly define the variables used in the study and indicate their significance.
• Provide a detailed explanation of the validation stages:
• Specify the datasets used for testing the model.
• Include the performance metrics to determine the accuracy of model

Validity of the findings

1. Results Interpretation

While the justification of presented results is satisfactory, however, I suggest refining this part after adjusting the methodology part as per my suggestions
Consider methodological limitations and acknowledge any uncertainties.

2. Literature Integration

Incorporate recent studies to validate your findings and strengthen the discussion. -

Additional comments

Ensure the paper adheres to standard academic formatting and referencing guidelines.

Improve the quality of figures by providing HD outlook, clear captions, and legends. At present these give a bit of a blurry look.

Thorough Proofreading of the manuscript is suggested to correct any grammatical or typographical errors, which are many at present
Overall a good effort. Just address minor observations and your are good to go

Reviewer 2 ·

Basic reporting

Language is clear and text well organized.
References are adequate and demonstrate the author as a solid base of information on the addressed topic. In general Figures, tables and references are correct.

Experimental design

Research topic is relevant, original and meaningful, and within the scope of the journal. But…
Applied methods are focused on the statistical analysis of the storage volumes in 10 reservoirs. All calculations are based on a statical evaluation of a long period of daily storage data (year 200 to 2025) for each of the 10 reservoirs. Data gaps have been filled and storage records where normalized (0 to 1) to account for storage capacity differences between the reservoirs.
Applied statistics is adequate, well explained and supported and the analysis of the results also seems sound and correct. And the conclusion by the author that reservoirs are unevenly managed in terms of stored volumes appears to be relevant.
But then multiple conclusions are presented without any further analysis. Based on the statistical analysis of storage the author concludes that reservoir management is not adequate and from this point on suggests multiple management solutions (presented as discussed).
The central problem is that no other information regarding the reservoirs is presented: storage capacity, drainage areas, water uses, inflow and water uses balance, residence time… yes, storage is normalized but one cannot judge reservoir management only based on storage variations. As an example, a small reservoir tends to have high storage variations due to its probable short residence time; bigger reservoirs must be more resilient as storage may be difficult to be recovered in a short time.
Analysing reservoir performance and even more, management, requires much more information them provided by the author. And there are no indications other essential variables have been accounted for.
Although applied methods are correct, performed analysis is to short and hardly enough for one to draw conclusions as the ones suggested by the author.
Some further details should also be considered:
1. In the abstract, line 28: Time series analysis show the seasonal fluctuations…. May be related to the effects of climate change”. This hypothesis/conclusion cannot be drawn only based on storage fluctuations! What about water use? What about drainage basin intersections? Conclusion is to general or at least based on evidence.
2. Also in the abstract, described strategies are common and general solutions in the topic of water resources management. No news here.
3. Line 34, the expression “underperforming dams” has no real basis on the analysis of the data as these do not demonstrate such underperformance. High storage fluctuations may suggest so but it is not enough.
4. Line 53: “Therefore, the occupancy rates of Istambul dams should be analysed by combining current hydrometeorological data with future climate scenarios”. The referenced authors (Zabalza et al) did not express this regarding Istambul dams, as their study area is in Spain. This reference is thus not correct.
5. Line 64: the same mistake as Park and Kim did not mention Istambul dams in their own study although they did conclude that modeling should be used. But they did not mention Istambul dams as the author, by mistake, suggests.
6. Line 83-86. Same mistake, referencing conclusions on Istambul dams…
7. Line 108-110. Paragraph must be reviewwd.
8. Line 186 – First time pdf is mentioned?
9. Line 213. This paragraph ignores essential data such as size, water use, drainage areas… Geography by itself does not explain what the author suggests: performance – location relation.
10. Line 310. The author describes influences on dam’s storage from variables that where not account for within the study: precipitation, evaporation
11. Line 336-342. All suggested measures, except maybe the first, are general and could be identified without any study. For instance, reducing water losses…? This is just one of the major problems in all the world.
12. Line 377. “Smart water management”: all general suggestions for which the only information required would be to know if such approaches are indeed or not being implemented – and this question is not addressed with the study.

Validity of the findings

Developed method are correct but are not enough to support all the conclusions (if any). Suggested improvement in water management are general and applicable to most systems.

The study is relevant but is very general and lacks essential information. Analysis is simplified to the maximum and conclusion are drawn from insufficient data – maybe for that reason conclusion are general.

It is my opinion that the paper should be rejected or at least subject to major improvements.

Annotated reviews are not available for download in order to protect the identity of reviewers who chose to remain anonymous.

---

## Round 0.2 · accepted · Accept

I am pleased to inform you that your manuscript entitled "Analysis of the occupancy rates of İstanbul Dams and optimum water management strategies against climate change effects" has been accepted for publication in PeerJ.

Following a thorough peer-review process and your subsequent revisions, the reviewers and I are satisfied that the manuscript meets PeerJ’s standards for scientific and methodological rigor, clarity, and relevance. Your work provides valuable insights into water resource management under climate change scenarios, with particular significance for regional and global discussions on sustainable infrastructure planning.

The PeerJ production team will now guide your manuscript through the final stages of copyediting, typesetting, and proof generation. You will receive a proof version for review prior to its online publication.

On behalf of PeerJ, I would like to thank you for choosing our journal as the venue for your research and for your constructive engagement throughout the review process. I also extend my appreciation for your contribution to the advancement of knowledge in this important field.

Congratulations on the acceptance of your work. I look forward to seeing your article published soon.

Best regards,

Dr. Armando Sunny

·

Basic reporting

The revisions are satisfactory

Experimental design

The revisions are satisfactory

Validity of the findings

The revisions are satisfactory

Additional comments

The author/ (S) have done a good job in making revisions to the manuscript as per my suggestions. I feel it can be accepted for publishing in current form.